# Effect of pH and Shear on Heat-Induced Changes in Milk Protein Concentrate Suspensions

**DOI:** 10.3390/foods13101517

**Published:** 2024-05-13

**Authors:** Anushka Mediwaththe, Thom Huppertz, Jayani Chandrapala, Todor Vasiljevic

**Affiliations:** 1Advanced Food Systems Research Unit, Institute of Sustainable Industries & Liveable Cities, College of Sports, Health and Engineering, Victoria University, Werribee Campus, Werribee, VIC 3030, Australia; anushka.mediwaththe@live.vu.edu.au (A.M.); thom.huppertz@frieslandcampina.com (T.H.); 2FrieslandCampina, P.O. Box 1551, 3800 BN Amersfoort, The Netherlands; 3Food Quality and Design Group, Wageningen University & Research, 6708 Wageningen, The Netherlands; 4School of Science, RMIT University, Bundoora, VIC 3083, Australia; jayani.chandrapala@rmit.edu.au

**Keywords:** dairy, milk protein concentrate, shear effect, temperature, pH variation

## Abstract

The effect of shear on heat-induced changes in milk protein concentrate suspensions was examined at different pH levels, revealing novel insights into micellar dissociation and protein aggregation dynamics. Milk protein concentrate suspensions, adjusted to pH of 6.1, 6.4, 6.8, or 7.5, underwent combined heat (90 °C for 5 min or 121 °C for 2.6 min) and shear (0, 100, or 1000 s^−1^) treatment. The fragmentation of protein aggregates induced by shear was evident in the control MPC suspensions at pH 6.8, irrespective of the temperature. At pH 7.5, shear increased the heat-induced micellar dissociation. This effect was particularly pronounced at 121 °C and 1000 s^−1^, resulting in reduced particle size and an elevated concentration of κ-casein (κ-CN) in the non-sedimentable phase. At pH 6.1 or 6.4, shear effects were dependent on sample pH, thereby modifying electrostatic interactions and the extent of whey protein association with the micelles. At pH 6.1, shear promoted heat-induced aggregation, evidenced by an increase in particle size and a significant decline in both whey proteins and caseins in the non-sedimentable phase. At pH 6.4, shear-induced fragmentation of aggregates was observed, prominently due to comparatively higher electrostatic repulsions and fewer protein interactions. The influence of shear on heat-induced changes was considerably impacted by initial pH.

## 1. Introduction

Milk protein concentrates (MPCs) are typically produced by the ultrafiltration of pasteurized skim milk, resulting in a concentrated mixture rich in casein and whey proteins, which is then typically dried to an MPC powder. During this process, the permeate stream, which contains lactose, water, and (soluble) milk salts, is removed [1]. While the protein-to-total-solids ratio is increased in the MPC, the casein-to-whey-protein ratio remains similar to that of skim milk [1,2]. These characteristics of MPC make it suitable for high-protein dairy beverages, which often undergo UHT (ultra-high temperature) and retort sterilization to enhance their shelf life [3,4]. 

The stability of milk proteins and their susceptibility to denaturation and aggregation during thermal processing depends on various factors, including heating temperature and time, mechanical forces like shear, pH, and the relative abundance of proteins and minerals [5,6]. Changes in pH can influence attractive and repulsive electrostatic interactions between protein molecules, thereby also impacting the structure of the casein micelles. The modification of milk pH prior to heat treatment affects the degree of association of denatured whey proteins with micelles [7]. After heat treatment of milk with a pH < 6.8, a substantial proportion of the whey protein is found attached to the casein micelles. After heating milk with a pH > 6.8, whey protein aggregates are largely found in the serum, upon association with κ-CN and dissociation from the casein micelles. However, heat-induced dissociation of micellar κ-CN in MPC occurs to a lesser extent than in milk, and only at higher pH values; it has been observed that at a pH > 6.8, the destabilizing effects of increased calcium ion activity due to the higher protein content in MPC powders ranging from MPC35 to MPC90 are partially offset by a decrease in heat-induced κ-CN dissociation [8,9].

While pH adjustment seems an obvious strategy to manipulate the heat stability of MPC, combining it with the simultaneous application of shear may create certain properties that would modify their structural characteristics, eventually leading to more pronounced effects. High shear could lead to structural changes in the casein micelle, e.g., either elongation along with the fluid drag or swelling, which would potentially improve the association of micellar caseins with the outer environment [10]. The intricate interplay between pH and shear dynamics was especially evident at low pH; for example, at pH 4.6 and particularly at pH 2.0, a notable reduction in the size of casein aggregates was observed with high shearing [11]. In the serum, these caseins were present as relatively small, soluble aggregates, primarily composed of α_S_- and β-caseins. The composition of these aggregates changed with the application of shear, suggesting the occurrence of orthokinetic aggregation and fragmentation [11]. Apart from that, whey proteins are also known to undergo shear-induced structural transformations, resulting in the unfolding and propagation of interactions that lead to aggregation within the milk system [5].

These aforementioned findings suggest that shear application in conjunction with other factors has the potential to modulate the structural properties of MPC dispersions. While previous research has explored the effects of pH and shear on MPCs, there remains a significant gap in the literature regarding the combined influence of these factors during heat treatment, particularly at the pH levels commonly encountered in various dairy applications ranging from pH 6.1 to 7.5. Therefore, studying the combined effects of heat and shear in a pH-modified environment can provide valuable insights into protein–protein interactions and their potential implications. The present study focused on investigating the influence of pH (6.1, 6.4, 6.8, or 7.5) and shear (0, 100, or 1000 s^−1^) on heat-induced (90 °C for 5 min and 121 °C for 2.6 min) changes in milk proteins in 8% MPC suspensions. 

## 2. Materials and Methods

### 2.1. Materials

MPC powder was obtained from Fonterra Co-operative (Palmerston North, New Zealand). The composition of the MPC powder, according to the manufacturer’s specifications, was 81.0% (*w*/*w*) total protein, 1.6% (*w*/*w*) fat, 5.5% (*w*/*w*) carbohydrate, and 7.2% (*w*/*w*) ash. Chemicals for analysis were purchased from Sigma-Aldrich Pty Ltd., based in Castle Hill, NSW, Australia, and ultrapure water (Milli-Q water, Merck Millipore, Bayswater, VIC, Australia) was consistently used throughout the study.

### 2.2. Sample Preparation and Treatment

MPC suspensions were prepared by reconstituting MPC powder in Milli-Q water, achieving a pH of 6.8 after reconstitution. The solutions were stirred continuously at 50 °C for 1 h to ensure complete solubilization of the powder, followed by overnight storage at 4 °C for further hydration [12]. The next day, samples were allowed to equilibrate at 25 °C for 1 h and the pH of the MPC suspensions was adjusted to 6.1, 6.4, 6.8, or 7.5 with concentrated HCl or NaOH. The final volume was adjusted using Milli-Q water to obtain a final protein concentration of 8% (*w*/*w*).

The MPC suspensions were subjected to heat treatment of 90 °C for 5 min or 121 °C for 2.6 min. A constant shear rate of either 100 or 1000 s^−1^ was maintained inside a pressure cell (CC25/PR-150) of a rheometer (Physica MCR 301 series, Anton Paar GmbH, Ostfildern-Scharnhausen, Germany). A constant pressure of 250 kPa was maintained throughout the experiments. This pressure is crucial for eliminating air bubbles, enhancing sample-to-measuring surface contact, stabilizing the sample during testing, ensuring consistent experimental conditions, and preventing evaporation when heating the samples, thereby facilitating accurate and reproducible rheological measurements across various samples. Samples subjected to heating under two temperatures were heated at a rate of 5 °C min^−1^ to the targeted temperature and held there for the required time and cooled at a rate of 5 °C min^−1^.

The pH of each sample was measured immediately after treatment using a pH meter (WTW Inolab pH 720, Weilheim, Germany). 

After processing, the samples were separated via sequential centrifugation using a Beckman L-70 Ultracentrifuge equipped with a Type 70.ITI rotor (Beckman Instruments, Inc., Brea, CA, USA) at 20 °C at 5700 or 75,940× *g* for 1 h, following a method previously described [13]. The resulting supernatants were collected for protein profile analysis. The protein fraction in the supernatants after centrifugation at 75,940 g was classified as “non-sedimentable”. The proteins that were sedimented following the 5700 g centrifugation were classified as “aggregated”.

### 2.3. Particle Size and Zeta Potential Measurements

Particle size and zeta potential measurements were conducted immediately after each treatment as described previously [14]. The measurements were performed using a Zetasizer (Zetasizer Nano ZS, Malvern Instruments, Malvern, UK). Prior to the measurements, the treated samples were diluted 1000 times using skim milk ultra-filtrate. In the calculations, the refractive indexes of 1.57 and 1.38 were used for MPC and skim milk ultra-filtrate, respectively.

### 2.4. Fourier Transform Infrared (FTIR) Analysis

The assessment of changes in the secondary structure of proteins was conducted using an FTIR spectrometer (PerkinElmer Frontier FTIR Spectrometer, Boston, MA, USA) as described previously [14]. FTIR spectra were acquired at room temperature (~20 °C) within 10 min after each treatment. Each spectrum was an average of 16 scans with a resolution of 4 cm^−1^ after subtracting the background. To enhance the resolution for qualitative analysis, the second derivative of all FTIR spectra was obtained within the broad amide I region of 1700–1600 cm^−1^. The obtained spectra were processed using Fourier self-deconvolution (FSD) and baseline correction with Origin Student 2019b software (Origin Lab Corporation, Northampton, MA, USA). The areas of the prominent peaks assigned to specific secondary structures were summed up and divided by the total area, resulting in the identification of five major peak areas corresponding to protein secondary structures, including intramolecular β sheets (1637–1615 cm^−1^), aggregated β sheets (1700–1682 cm^−1^), random coils (1645–1638 cm^−1^), α-helices (1664–1646 cm^−1^), and β turns (1681–1665 cm^−1^). The obtained results were then subjected to statistical analysis following the guidelines outlined in Section 2.6.

### 2.5. Reverse Phase-High Performance Liquid Chromatography Analysis

Whole samples and supernatants of each centrifuged MPC suspension were analysed for the content of α_S1_-casein (α_S1_-CN), α_S2_-casein (α_S2_-CN), β-casein (β-CN), κ-casein (κ-CN), α-lactalbumin (α-LA), and β-lactoglobulin (β-LG) by RP-HPLC using a Zorbax 300SB-C8 RP-HPLC column (silica-based packing, 3.5 micron, 300A, Agilent Technologies Inc., Mulgrave VIC, Australia) as the stationary phase. Water (mobile phase A) and acetonitrile (mobile phase B) solutions both containing 0.1% (*v*/*v*) trifluoroacetic acid (TFA) were used as mobile phases. A gradient elution was run at a constant flow rate of 0.8 mL/min as described previously [15].

### 2.6. Statistical Analysis

Statistical analysis was conducted using IBM SPSS Statistics software (version 28.0.1.0, IBM Corp., Armonk, NY, USA) employing a general linear model (GLM) approach. The study was arranged as a randomized block, full factorial design with pH as the main plot, while the subplots were temperature/time combinations and shearing. This block was replicated at least twice, with each replication consisting of three sub-samplings. The level of significance was set at *p* ≤ 0.05. 

## 3. Results

### 3.1. Particle Size Distribution and Zeta Potential of MPC Suspensions

In unheated MPC suspensions, the largest particles (~235 nm) were observed in the sample at pH 6.1 and the lowest particle size at pH 6.8 (~207 nm) (Figure 1 and Table 1). 

Heating the MPC suspensions with initial pH values of 6.1 or 6.4 led to an increase in particle size and these were larger at pH 6.1 than at pH 6.4 (Figure 1 and Table 1); the extent of these increases was far larger than what would be expected from whey protein denaturation [16,17] and suggests it to be due to casein micelle aggregation. In contrast, at higher pH (≥pH 6.8), heating consistently reduced particle size, indicating a tendency towards the formation of smaller aggregates or the dissociation of casein micelles. Heating the MPC suspensions without shear at pH 6.8 caused a reduction in particle size, from ~207 nm at 20 °C, to ~182 nm at 90 °C, and ~179 nm at 121 °C (Figure 1 and Table 1). This decrease may be attributed to the heat-induced dissociation of casein micelles or the formation of smaller whey protein aggregates [18]. A similar trend was observed in the MPC suspension with a pH of 7.5, showing a reduction in particle size from ~214 nm at 20 °C, to ~196 and ~184 nm after heating at 90 and 121 °C, respectively. This decrease can be mainly attributed to the formation of smaller soluble aggregates [19]. 

The application of shear, in conjunction with heat, further influenced these trends. At pH 6.1, the combined application of heat and shear resulted in an increase in particle size, particularly at 121 °C and 1000 s^−1^ (Table 1). This is in contrast to the behaviour observed at pH 6.4, where similar conditions led to a significant decrease in particle size with an increasing shear rate. At pH 6.8, the combined application of heat and shear did not change the particle size. However, at pH 7.5, a consistent trend of particle size reduction under the influence of heat and shear was observed, with the smallest particle size observed at a shear rate of 1000 s^−1^ at 121 °C. 

At 20 °C, a zeta potential of ~−18 mV was observed in MPC suspensions at both pH 6.1 and 6.4. Increasing the pH to 6.8 and 7.5 led to a rise in the zeta potentials of ~−21 mV and −25 mV, respectively, highlighting a pH-dependent trend in the increase in negative zeta potential. However, heating did not notably affect the zeta potential of the MPC suspensions at different pH levels (Table 2). 

The application of shear during heating significantly (*p* < 0.05) influenced the zeta potential, particularly at lower pH values. For example, when MPC suspensions at pH 6.1 were heated at 121 °C and sheared at 1000 s^−1^, the zeta potential changed from ~−20 mV in the absence of shear to ~−17 mV. Conversely, at 90 °C, the zeta potential at pH 6.1 remained consistent across different shear rates, indicating that, at this temperature, shear forces did not significantly influence the zeta potential. In contrast, at pH 6.4, the combined application of heat and shear resulted in an increased zeta potential at both 90 °C and 121 °C compared to the suspensions at 20 °C. The increase was more pronounced at 121 °C, where the zeta potential rose to ~−20.5 mV at 1000 s^−1^, from ~−18.6 mV at 20 °C. The combined application of heat and shear did not result in any significant changes in the zeta potential of MPC suspensions at pH 6.8 and 7.5 (Table 2).

### 3.2. Partitioning of Proteins in MPC Suspensions

#### 3.2.1. Changes in Protein Content within the Non-Sedimentable Fraction

In the unheated sample at pH 6.8, the non-sedimentable (75,490× *g* for 60 min at 20 °C) fraction contained ~15% of total α_S1_-CN, ~20% of total α_S2_-CN, ~20% of total β-CN, and ~28% of total κ-CN (Table 3), as well as most of the β-LG (~97%) and α-LA (~98%) (Table 3). 

Lowering the pH resulted in a notable decrease in the content of both caseins and whey proteins in the non-sedimentable fraction. Specifically, at pH 6.4 and further at 6.1, the κ-CN content decreased to ~13% of its total, alongside reductions in α_S1_-CN, α_S2_-CN, and β-CN levels. Moreover, this pH adjustment led to a more pronounced decrease in β-LG and α-LA levels, with the most significant reduction occurring at pH 6.1, where β-LG and α-LA levels decreased by >10% compared to those at pH 6.4. On the other hand, an increase in pH to 7.5 resulted in an increase in non-sedimentable κ-CN, up to ~50% of its total, as well as increased levels of non-sedimentable α_S1_-CN, α_S2_-CN, and β-CN (Table 3).

At pH 6.8, heat treatment alone resulted in an increase in levels of non-sedimentable κ-CN, reaching ~42% of the total κ-CN at 121 °C (Table 3). The levels of non-sedimentable β-CN, α_S1_-CN, and α_S2_-CN also increased. Simultaneously, both β-LG and α-LA declined with β-LG showing a more pronounced decrease by ~41% at 121 °C. At pH < 6.8, heating further reduced both the casein and whey protein levels in the non-sedimentable fraction compared to non-heated suspensions. 

At pH 6.4 or 6.1, heating at 121 °C without shear resulted in a notable reduction in non-sedimentable κ-CN, while the other caseins were not found in the non-sedimentable phase. This was also accompanied by a decrease in non-sedimentable β-LG and α-LA (Table 3). At pH 7.5, heat treatment further increased non-sedimentable κ-CN levels, up to ~63% of total κ-CN at 121 °C. Additionally, an increase in β-CN (~12% of the total at 121 °C) and slight increases in α_S1_-CN and α_S2_-CN (~6% at 121 °C) were also noted at this pH. Non-sedimentable β-LG and α-LA also declined at this pH with an effect more pronounced for β-LG.

The combined application of heat and shear, along with an increase in pH, notably enhanced the levels of all caseins, particularly κ-CN, in the non-sedimentable fraction. For example, at pH 6.8, non-sedimentable κ-CN increased, e.g., by ~15% of total κ-CN at 121 °C when the shear rate was changed from 0 s^−1^ to 1000 s^−1^ (Table 3). Levels of non-sedimentable α_S1_-CN, α_S2_-CN, and β-CN also showed direct shear dependency as their concentration rose at both temperatures and pH 6.8, with a more notable rise observed in β-CN, reaching up to ~57% of total β-CN (Table 3). Furthermore, at these conditions at pH 6.8, non-sedimentable β-LG and α-LA also increased by ~22% and ~28% of their totals, respectively, at 121 °C and a shear rate of 1000 s^−1^ (Table 3). 

The combined application of heat and shear elevated the levels of both casein and whey proteins in the non-sedimentable phase as the pH increased. At pH 6.1, the level of caseins within the non-sedimentable fraction was unaffected by the combined application of heat and shear. However, the applied shear resulted in a decrease in both β-LG and α-LA (Table 3). At pH 6.4, the combined application of heat and shear resulted in an increase in non-sedimentable κ-CN content at both temperatures (Table 3), e.g., by ~29% at 121 °C and a shear rate of 1000 s^−1^. No distinct trend could be observed for α_S1_-CN while both α_S2_-CN and β-CN experienced slight increases. Under these conditions, both β-LG and α-LA concentrations in the non-sedimentable fraction increased. The impact was more prominent on β-LG with a 24% increase at 121 °C and at 1000 s^−1^ compared to 16% in α-LA under the same conditions. 

At pH 7.5, the combined application of heat and shear significantly increased the κ-CN content, reaching ~86% of the total at 121 °C and a shear rate of 1000 s^−1^ (Table 3). Additionally, the levels of non-sedimentable α_S1_-CN, α_S2_-CN, and β-CN also increased. For instance, at 121 °C and a shear rate of 1000 s^−1^, the concentration of α_S1_-CN, α_S2_-CN, and β-CN increased by ~9, ~16, and ~18%, respectively. Furthermore, both β-LG and α-LA levels were also elevated. Specifically, β-LG and α-LA contents increased by ~12 and ~14%, respectively, at 121 °C and a shear rate of 1000 s^−1^.

#### 3.2.2. Changes in the Content of Aggregated Protein

In the unheated sample at pH 6.8, the aggregated fraction, defined as the fraction that had sedimented after centrifugation at 5700× *g* for 60 min at 20 °C, contained ~5% of total α_S1_-CN, ~7% of total α_S2_-CN, ~8% of total β-casein, and ~10% of total κ-CN (Table 3), and very little β-LG (~2%) and α-LA (~1%) (Table 4). Lowering the pH increased both caseins and whey proteins within the aggregated fraction with a pronounced increase observed at pH 6.1. Decreasing the pH to 6.4 or 6.1 resulted in a substantial increase in sedimented κ-CN, reaching ~36% and ~53%, respectively (Table 4).

In addition, the levels of α_S1_-CN, α_S2_-CN, and β-CN appeared to be pH-dependent as their content in the aggregated fraction was greatest at pH 6.1, up to ~80% of their total. Lowering the pH to 6.4 or 6.1 also resulted also in a more pronounced increase in β-LG and α-LA levels in the aggregated fractions. Increasing pH to 7.5 did not affect the levels of caseins in this fraction noticeably, with only a slight increase observed in both β-LG and α-LA.

Heat treatment without shear increased κ-CN levels in the aggregated fraction (e.g., to ~36% of the total after heat treatment at 121 °C) at pH 6.8 (Table 4). This was also accompanied by a rise in the levels of aggregated β-CN, α_S1_-CN, and α_S2_-CN (Table 4). Both β-LG and α-LA levels in the aggregated fraction were also greater, with a comparatively greater increase observed for β-LG, especially at 121 °C, where ~25% of the total β-LG and 20% of the total α-LA were observed. Reducing the pH to 6.4 or 6.1 led to a notable increase in all caseins in the aggregated fraction. Additionally, the levels of β-LG and α-LA also increased, particularly more at pH 6.1, reaching ~52% and ~46%, respectively (Table 4). At pH 7.5, heating of MPC suspensions did not change the levels of κ-CN and α_S1_-CN in the aggregated fraction, but β-CN and α_S2_-CN levels increased by ~12% and ~6%, respectively, at 121 °C. This pH also affected the behaviour of α-LA and β-LG, with only β-LG rising up to ~17% at 121 °C in the aggregated fraction (Table 4).

The combined application of heat and shear showed that the protein composition of the aggregated fraction was clearly pH dependent—higher pH at heating and shearing resulted in less incorporation of caseins and whey proteins into the aggregates. For example, a small proportion of the total caseins was incorporated at pH 7.5 in comparison to those at other pH levels (Table 4). κ-CN appeared to be affected the most as its levels in the aggregates rose as the pH was lowered. In general, higher shear reduced the levels of caseins in the aggregates during heating (Table 4). However, this trend was somewhat reversed when pH was reduced to 6.1. Most caseins were incorporated in these aggregates, even in the absence of shear at this pH, and shear had no observable effect. The level of κ-CN increased from that without shear, indicating greater incorporation of this protein under the shear (Table 4). This, however, was the opposite to its levels at pH 6.4 as the level of κ-CN declined upon shearing indicating that electrostatic interactions may have governed some of the aggregation that could have been manipulated by shearing to a certain effect.

β-LG and α-LA followed a similar pattern to that of κ-CN as their levels in the aggregated fraction were clearly affected by pH and shear (Table 4). Their incorporation was minimal at pH 7.5 and, even then, it was further reduced by greater shearing. As pH was adjusted to 6.8 or 6.4, more and more of these proteins appeared to be incorporated into the aggregates. In these cases, the extent of shear was inversely related as it reduced their levels substantially at the greater shear, almost by 60% in comparison to those in the absence of shear. This effect, however, was reversed when the samples were heated and sheared at pH 6.1. The levels of β-LG and α-LA in this case increased by an average of 42–44% in comparison to these in the aggregates obtained under quiescent conditions (Table 4).

### 3.3. Conformational Properties of Proteins in MPC Suspensions

The proportion of secondary structures in the unheated samples depended on pH (Table 5). The secondary structure in the control unheated MPC suspensions was mainly intramolecular β-sheets accounting for ~45% of the total (Table 5). Adjusting the pH to 6.4 or 6.1 at 20 °C reduced the intramolecular β-sheets and α-helical structures with a more pronounced effect at pH 6.1 (Figure 2 and Table 5). 

Increasing pH to 7.5 led to a decrease in intramolecular β-sheets, up to ~40%, likely due to the disruption of native conformation of whey proteins and an increase in random structures due to changes in protein interactions, and this led to destabilisation of the micellar structure [20].

In the MPC suspensions at pH 6.8, heating resulted in a pronounced decrease in intramolecular β-sheet structures (Figure 2 and Table 5) combined with an increase in β-turn structures (Figure 2). Heating MPC suspensions without shear at pH 6.4 or 6.1 further decreased intramolecular β-sheets and α-helix structures (Table 5). At pH 7.5, heating resulted in a further decline in the content of intramolecular β-sheets and β-turns, accompanied by a prominent increase in the content of random structures. 

The concurrent application of heat and shear to MPC suspensions at pH 6.8 further reduced the intramolecular β-sheet structures while elevating the presence of β-turns. Additionally, a decrease in α-helix content and a simultaneous rise in random coils were observed (Figure 2 and Table 5). Moreover, the aggregated β-sheet structures also exhibited a decrease compared to heated dispersions. In the MPC suspensions with pH adjusted to pH 6.4 and 6.1, a shear-dependent reduction in intramolecular β-sheets and α-helical content was observed at both temperatures (Figure 2 and Table 5). At pH 7.5, the combined application of heat and shear decreased the content of α-helix structures compared to heated suspensions, with a more pronounced effect at 121 °C and at 1000 s^−1^.

## 4. Discussion

Adjusting the pH of the MPC suspensions before heat treatment significantly influenced the association between denatured whey proteins and casein micelles. Lowering the pH < 6.8 and heating promotes the formation of aggregates involving caseins and whey proteins. Particularly, at these lower pH levels, whey protein interactions with the casein micelle surface during heating lead to the formation of larger particles. Moreover, the relatively high amount of whey proteins associated with micelles may have facilitated the aggregation of casein particles through cross-linking of whey proteins bound to micelles [21]. Additionally, the reduction in pH and the altered dynamic equilibrium of minerals, particularly calcium and phosphate, between the serum and MCP nanoclusters caused by heating, lead to micellar destabilization. This results in the release of κ-CN, which makes the micelles more susceptible to aggregation. This is accompanied by a reduction in α-helical structures and β-sheets, linked to the disruption of hydrogen bonds and a less orderly arrangement of protein elements [20]. The pH of a solution significantly affects the protein behaviour, particularly through its relation to the protein’s isoelectric point (pI). The pI represents the pH at which a protein carries no net charge. For caseins, with a pI around 4.6, the proximity of the solution’s pH to the pI influences solubility and conformational changes, affecting interactions with other molecules. This correlation between casein’s pI and solution pH directly affects micellar stability and aggregation tendencies [11]. Therefore, a more acidic environment is likely to enhance electrostatic interactions among proteins by charge neutralization. This, in turn, can impact the stability of hydrogen bonds, leading to a significant reduction in both intramolecular β-sheets and α-helical content as observed in FTIR analysis (Figure 2 and Table 5) [20]. 

Adjusting pH to 7.5 leads to the loss and expansion of structures in the micelles, attributed to the higher surface potential with an increased susceptibility to disruption (Table 2) [22,23]. Several factors have been identified as contributing to the dissociation of casein micelles. These include a decrease in hydrophobic interactions among the caseins, increased electrostatic repulsions, and alterations in the mineral equilibrium involving calcium and phosphate [24,25,26]. In addition, a decrease in pH due to the liberation of hydrogen ions and changes to the dynamic equilibrium of minerals between colloidal and soluble phases with the heating considerably contributed to the destabilization of the micellar structure. Such destabilization aids in the dissociation and subsequent release of caseins into the serum, which increases the proportion of caseins in the soluble phase [27]. As a result of micellar dissociation, a reduction in particle size was observed in the MPC suspensions at pH 7.5 upon heating (Table 1) [7]. In addition, the altered charge distribution influences the stability of hydrogen bonds, manifesting a decline in β-sheets, β-turns, and α-helical structures while increasing random structures, signifying a less orderly arrangement of protein elements (Figure 2) [20].

The application of shear alters the dynamics of aggregation, shifting from perikinetic to orthokinetic aggregation. This transition involves particle collisions occurring within the flow streamlines [28]. Although the flow intensifies the frequency of collisions, it simultaneously introduces the potential for fracture of aggregates [29]. In the presence of shear, the growth dynamics and size of protein complexes are influenced by the interplay of shear-induced growth and shear-mediated breakage mechanisms [30]. In the control MPC suspensions at pH 6.8, prominent shear-induced fragmentation of protein aggregates was evident at both temperatures. This was due to an increase in all caseins and whey proteins in the non-sedimentable fraction, while their levels in the aggregated fraction were reduced (Table 3 and Table 4).

At pH 6.1, shear-induced collisions between particles may enhance aggregation compared to pH 6.4 (Figure 1). While shear consistently promotes particle collisions, the pH of the environment determines their propensity to stick together, with lower pH values favouring increased aggregation. As a result, in suspensions with a pH adjusted to 6.1, a more noticeable decrease in the levels of both caseins and whey proteins in the non-sedimentable fraction was observed compared to suspensions at pH 6.4 under the combined application of heat and shear (Table 3 and Table 4). On the other hand, higher electrostatic repulsion and less association of whey proteins with micelles at pH 6.4 compared to pH 6.1 may result in lower aggregation, thereby promoting prominent fragmentation (Figure 1 and Table 1). These changes in the charge distribution among proteins might have prompted conformational adjustments in proteins, resulting in a decline in intramolecular β-sheet content (Figure 2 and Table 5). The aggregation process itself could have induced alterations in protein conformation, thereby contributing to the observed reduction in β-sheet content. Protein aggregation may involve the formation of turns and loops in protein structures, elucidating the observed increase in β-turns at pH 6.1 (Table 5).

In the MPC suspensions with pH adjusted to 7.5, the application of combined heat and shear increased the casein levels in the non-sedimentable phase, with a more pronounced effect observed at 121 °C (Table 3 and Table 4). The fluid drag induced by shear flow may contribute to the structural transformations into elongated shapes and the exhibition of complex behaviour. For instance, at high concentrations, casein micelles can behave like soft spheres, deforming and aligning in the direction of the flow at elevated shear rates [10]. In addition to the looser and expanded structure of the micelles, weakened hydrophobic interactions resulting from high temperatures under the influence of pH make them susceptible to destabilization affecting the overall balance between interactions. The increased collision rate with the rise in shear leads to further disruption of micelles, causing the release of more κ-CN into the non-sedimentable phase (Table 3). These shear-induced mechanical forces disrupt the secondary structure of milk proteins, particularly reducing α-helical structures, through protein deformation, unfolding, and reorientation when subjected to shear flow as observed previously [5] (Table 5).

## 5. Conclusions

In conclusion, the interplay between shear and pH in MPC suspensions reveals a complex dynamic of protein interactions and structural alterations. At lower pH levels, such as pH 6.1, the application of shear enhances particle collisions, leading to increased aggregation compared to those at pH 6.4. This suggests that pH influences the propensity of particles to adhere, with lower pH values favouring heightened aggregation under shear conditions. Conversely, at pH 6.4, higher electrostatic repulsion and less association of whey proteins with micelles result in less aggregation, promoting prominent fragmentation. A shear-dependent reduction in intramolecular β-sheets and α-helical content was observed at both temperatures at both pH levels suggesting alterations in protein conformation. Shear-induced fragmentation of protein aggregates was clearly noticeable in the control MPC suspensions at pH 6.8, regardless of the heating temperature. The combination of heat and shear applied to MPC suspensions at pH 6.8 resulted in a reduction in intramolecular β-sheet structures and an increase in β-turns. There was also a decrease in α-helix content and an increase in random coils. Additionally, aggregated β-sheet structures decreased compared to heated dispersions. Adjusting the pH to 7.5 yields loose and expanded micellar structures, rendering them more susceptible to shear-induced disruption. Shear, in combination with heat, intensifies collisions and disrupts micelles, leading to the release of more κ-CN into the non-sedimentable phase. At pH 7.5, the combined application of heat and shear decreased the content of α-helical elements compared to heated suspensions due to changes in protein structure.

Overall, the findings underscore the intricate relationship between shear, pH, and protein behaviour in MPC suspensions. This understanding aids dairy manufacturers in optimizing processing conditions for improved product stability, texture, and functionality. It also supports the development of processing techniques that preserve protein functionality and nutritional integrity in dairy products. Further research could delve deeper into elucidating the specific mechanisms underlying shear-induced alterations in protein structures and interactions across a broader range of pH conditions.

## Figures and Tables

**Figure 1 foods-13-01517-f001:**
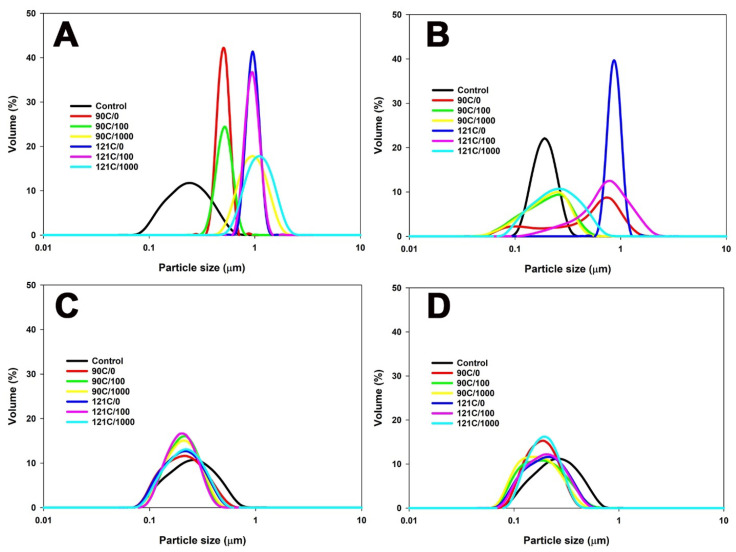
Particle size distribution of milk protein suspensions with pH adjusted to 6.1 (**A**), 6.4 (**B**), 6.8 (**C**), or 7.5 (**D**) and sheared at 100 s^−1^ or 1000 s^−1^ during heating at 90 °C for 5 min or 121 °C for 2.6 min. Please note that in the figure labels, ‘C’ represents degrees Celsius (°C), and the numerical value following the slash denotes shear rate per second (s^−1^). The true controls were assessed prior to heating without shear (0 s^−1^) at their respective pH.

**Figure 2 foods-13-01517-f002:**
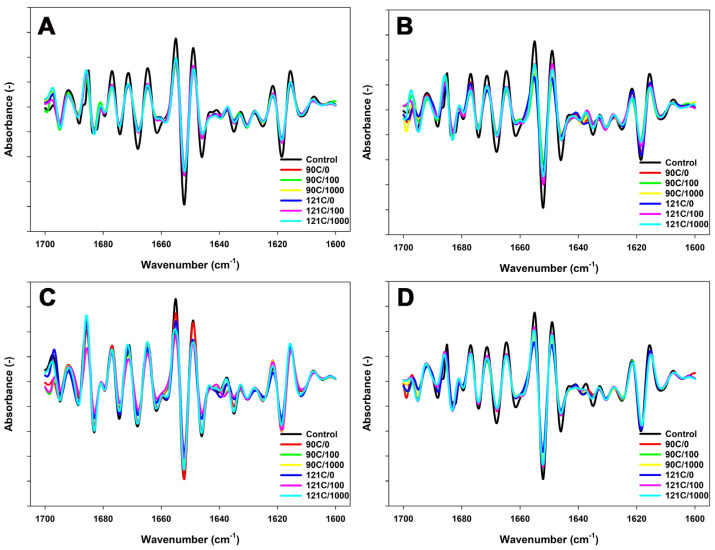
FTIR spectra (second derivative) for Amide I region of MPC dispersions with pH adjusted to 6.1 (**A**), 6.4 (**B**), 6.8 (**C**), or 7.5 (**D**) and sheared at 100 s^−1^ or 1000 s^−1^ during heating at 90 °C for 5 min or 121 °C for 2.6 min. Please note that in the figure labels. ‘C’ represents degrees Celsius (°C), and the numerical value following the slash denotes the applied shear rate per second (s^−1^). The true controls were assessed prior to heating without shear (0 s^−1^) at their respective pH.

**Table 1 foods-13-01517-t001:** Average particle size of MPC suspensions with pH adjusted to 6.1, 6.4, 6.8, or 7.5 and subjected to different temperatures (20, 90, or 121 °C) and shear rates (0, 100, or 1000 s^−1^).

pH	Temperature (°C)	Particle Size (nm)		
		0 s^−1^	100 s^−1^	1000 s^−1^
6.1	20	235 ^E^		
	90	714 ^Cc^	850 ^Bb^	1974 ^Ba^
	121	1632 ^Ac^	1842 ^Ab^	2132 ^Aa^
6.4	20	224 ^F^		
	90	384 ^Da^	330 ^Dc^	338 ^Db^
	121	1146 ^Ba^	796 ^Cb^	498 ^Cc^
6.8	20	207 ^H^		
	90	182 ^Ja^	180 ^Ea^	178 ^Ea^
	121	179 ^Ja^	178 ^Ea^	177 ^Ea^
7.5	20	214 ^G^		
	90	196 ^Ia^	174 ^Eb^	166 ^Fc^
	121	184 ^Ja^	180 ^Ea^	170 ^Fb^
SEM_pooled_ *		0.675		

Different uppercase letter superscripts indicate a significant difference among the means within a column, while lowercase superscripts signify a significant difference among the means within a raw; *p* < 0.05. * Pooled Standard Error of Mean.

**Table 2 foods-13-01517-t002:** Zeta potential of MPC suspensions with pH adjusted to 6.1, 6.4, 6.8, or 7.5 and subjected to different temperatures (20, 90, or 121 °C) and shear rates (0, 100, or 1000 s^−1^).

pH	Temperature (°C)	Zeta Potential (mV)		
		0 s^−1^	100 s^−1^	1000 s^−1^
6.1	20	−18.4 ^AB^		
	90	−19.0 ^ABa^	−19.0 ^BCa^	−20.2 ^ABa^
	121	−19.8 ^ABa^	−18.4 ^Aba^	−16.5 ^Ab^
6.4	20	−18.6 ^AB^		
	90	−17.8 ^Aa^	−17.5 ^Aa^	−18.9 ^ABb^
	121	−18.3 ^Aa^	−19.5 ^BCb^	−20.5 ^ABc^
6.8	20	−21.2 ^C^		
	90	−21.2 ^Ca^	−21.2 ^Da^	−21.1 ^Ba^
	121	−20.8 ^Ca^	−19.9 ^CDa^	−21.1 ^Ba^
7.5	20	−25.3 ^D^		
	90	−26.0 ^Da^	−26.4 ^Ea^	−26.8 ^Ca^
	121	−25.1 ^Da^	−25.9 ^Ea^	−25.5 ^Ca^
SEM_pooled_ *		0.419		

Different uppercase letter superscripts indicate a significant difference among the means within a column, while lowercase superscripts signify a significant difference among the means within a raw; *p* < 0.05. * Pooled Standard Error of Mean.

**Table 3 foods-13-01517-t003:** Proportion of α_s1_-CN, α_s2_-CN, β-CN, κ-CN, β-LG, and α-LA in the non-sedimentable fraction (75,490× *g* for 1 h at 20 °C), expressed as a percentage of their content in the original bulk unheated suspensions with pH adjusted to 6.8, subjected to different temperatures (20, 90, or 121 °C) and shear rates (0, 100, or 1000 s^−1^).

pH	Temperature (°C)	Shear Rate (s^−1^)	% Protein Content in Non-Sedimentable Fraction					
			α_s1_-CN	α_s2_-CN	β-CN	κ-CN	β-LG	α-LA
6.1	20	0	2.4 ^g^	3.6 ^h^	5.0 ^j^	12.5 ^k^	70.5 ^c^	76.0 ^fgh^
	90	0	0.0	0.0	0.0	6.0 ^m^	41.7 ^i^	52.4 ^kl^
	90	100	0.0	0.0	0.0	4.5 ^m^	30.0 ^j^	20 ^n^
	90	1000	0.0	0.0	0.0	3.0 ^m^	12.0 ^l^	11.1 ^o^
	121	0	0.0	0.0	0.0	4.3 ^m^	23.3 ^k^	39.4 ^m^
	121	100	0.0	0.0	0.0	4.7 ^m^	22.2 ^k^	16.7 ^n^
	121	1000	0.0	0.0	0.0	3.0 ^m^	20 ^k^	8.0 ^o^
6.4	20	0	4.3 ^g^	9.3 ^g^	8.3 ^ij^	12.5 ^k^	85.0 ^b^	81.3 ^de^
	90	0	0.0	1.7 ^h^	2.7 ^j^	12.1 ^k^	52.8 ^gh^	65.1 ^j^
	90	100	0.0	13.6 ^g^	8.6 ^ij^	40.5 ^hi^	57.3 ^fg^	71.8 ^hi^
	90	1000	0.0	10.7 ^g^	11.4 ^i^	62.5 ^cd^	68.6 ^c^	81.9 ^de^
	121	0	0.0	0.0	0.0	7.1 ^lm^	41.5 ^i^	57.5 ^k^
	121	100	0.0	0.0	0.0	25.0 ^j^	53.3 ^gh^	65.5 ^j^
	121	1000	0.0	0.0	0.0	36.4 ^i^	65.6 ^cde^	73.1 ^ghi^
6.8	20	0	15.0 ^f^	20.0 ^f^	20.0 ^h^	28.0 ^j^	97.0 ^a^	98.0 ^a^
	90	0	16.5 ^ef^	24.8 ^def^	28.7 ^fg^	36.2 ^i^	51.1 ^h^	68.0 ^ij^
	90	100	21.1 ^cde^	22.5 ^ef^	33.7 ^def^	46.7 ^fg^	66.3 ^cd^	76.1 ^fgh^
	90	1000	22.6 ^bcd^	25.0 ^d^	41.8 ^b^	63.0 ^cd^	70.0 ^c^	77.0 ^efg^
	121	0	20.5 ^cdef^	29.4 ^cd^	35.5 ^cde^	42.3 ^gh^	40.5 ^i^	49.6 ^l^
	121	100	23.5 ^bcd^	34.0 ^bc^	53.3 ^a^	56.5 ^e^	60.4 ^efg^	71.6 ^hi^
	121	1000	23.8 ^bcd^	34.4 ^bc^	57.0 ^a^	57.3 ^e^	62.9 ^def^	77.5 ^efg^
7.5	20	0	21.3 ^cde^	21.7 ^ef^	26.4 ^g^	49.5 ^f^	81.9 ^b^	90.5 ^b^
	90	0	19.8 ^def^	25.0 ^def^	31.0 ^efg^	58.2 ^de^	53.3 ^gh^	81.8 ^de^
	90	100	25.0 ^bc^	25.9 ^de^	40.0 ^bc^	65.6 ^c^	68.9 ^c^	85.4 ^bcd^
	90	1000	33.3 ^a^	38.7 ^b^	42.7 ^b^	73.7 ^b^	67.1 ^cd^	89.8 ^bc^
	121	0	26.8 ^b^	27.8 ^d^	38.8 ^bcd^	63.4 ^c^	42.9 ^i^	68.9 ^ij^
	121	100	22.2 ^bcd^	42.1 ^a^	54.7 ^a^	81.6 ^a^	51.7 ^h^	78.6 ^ef^
	121	1000	36.2 ^a^	43.9 ^a^	56.7 ^a^	85.9 ^a^	54.9 ^gh^	82.9 ^cde^
SEM_pooled_ *			0.143	0.034	0.210	0.085	0.120	0.097

Different lowercase letter superscripts indicate a significant difference among the means within a column (*p* < 0.05). * Pooled Standard Error of Mean.

**Table 4 foods-13-01517-t004:** Content of α_s1_-CN, α_s2_-CN, β-CN, κ-CN, β-LG, and α-LA in the sedimentable fraction (5700 g for 60 min at 20 °C) as a percentage of their content in the original bulk unheated suspensions with pH adjusted to 6.8, subjected to different temperatures (20, 90, or 121 °C) and shear rates (0, 100, or 1000 s^−1^).

pH	Temperature (°C)	Shear Rate (s^−1^)	% Protein Content in Sedimentable Fraction					
			α_s1_-CN	α_s2_-CN	β-CN	κ-CN	β-LG	α-LA
6.1	20	0	85.7 ^b^	81.0 ^c^	80.0 ^d^	52.5 ^d^	14.1 ^ij^	15.7 ^gh^
	90	0	99.8 ^a^	89.7 ^b^	85.0 ^c^	68.8 ^c^	43.3 ^d^	37.5 ^e^
	90	100	100.0 ^a^	98.5 ^a^	96.0 ^a^	73.2 ^c^	55.0 ^c^	55.0 ^c^
	90	1000	100.0 ^a^	99.2 ^a^	98.0 ^a^	80.0 ^b^	68.7 ^b^	61.1 ^b^
	121	0	98.5 ^a^	99.7 ^a^	90.0 ^b^	78.3 ^b^	52.2 ^c^	46.2 ^d^
	121	100	100.0 ^a^	100.0 ^a^	98.0 ^a^	80.0 ^b^	64.4 ^b^	58.3 ^b^
	121	1000	100.0 ^a^	100.0 ^a^	100.0 ^a^	85.0 ^a^	74.0 ^a^	67.0 ^a^
6.4	20	0	61.3 ^e^	41.9 ^f^	58.3 ^g^	36.3 ^e^	2.0 ^k^	4.4 ^jkl^
	90	0	75.7 ^d^	75.9 ^d^	64.8 ^f^	50.0 ^d^	22.3 ^fg^	11.8 ^hi^
	90	100	80.8 ^c^	72.7 ^d^	68.5 ^f^	24.3 ^gh^	17.4 ^ghi^	9.7 ^ij^
	90	1000	79.1 ^cd^	71.4 ^d^	70.5 ^ef^	12.5 ^i^	10.4 ^j^	5.5 ^jkl^
	121	0	98.7 ^a^	81.0 ^c^	70.0 ^ef^	73.8 ^c^	31.4 ^e^	27.5 ^f^
	121	100	100.0 ^a^	73.5 ^d^	73.6 ^e^	33.3 ^ef^	22.9 ^fg^	18.5 ^g^
	121	1000	100.0 ^a^	62.5 ^e^	75.0 ^e^	30 ^f^	11.1 ^j^	11.6 ^hi^
6.8	20	0	5.0 ^h^	7.0 ^l^	8.0 ^k^	10.0 ^ij^	2.0 ^k^	1.0 ^l^
	90	0	19.3 ^g^	23.9 ^gh^	18.4 ^i^	25.3 ^g^	21.1 ^fgh^	15.1 ^gh^
	90	100	7.8 ^h^	19.1 ^hi^	14.1 ^ij^	18.5 ^h^	10.5 ^j^	7.6 ^ijk^
	90	1000	5.4 ^h^	15.2 ^ij^	8.8 ^k^	10.0 ^ij^	10.0 ^j^	12.0 ^hi^
	121	0	25.1 ^f^	28.2 ^g^	25.6 ^h^	35.6 ^e^	24.8 ^f^	19.6 ^g^
	121	100	5.1 ^h^	5.3 ^l^	2.2 ^l^	23.5 ^gh^	8.8 ^j^	7.4 ^ijk^
	121	1000	3.2 ^i^	5.8 ^l^	1.1 ^l^	19.8 ^h^	9.3 ^j^	9.0 ^ij^
7.5	20	0	6.1 ^h^	9.6 ^k^	8.3 ^k^	10.8 ^ij^	4.2 ^k^	3.6 ^k^
	90	0	7.7 ^h^	14.3 ^ijk^	18.4 ^i^	13.8 ^i^	18.0 ^ghi^	3.4 ^k^
	90	100	4.5 ^h^	12.9 ^jk^	16.7 ^i^	2.1 ^k^	9.5 ^j^	2.4 ^k^
	90	1000	4.6 ^h^	9.9 ^kl^	10.6 ^jk^	4.0 ^k^	10.0 ^hi^	2.5 ^k^
	121	0	8.5 ^h^	16.5 ^ij^	25.5 ^h^	13.5 ^i^	16.5 ^hi^	1.3 ^l^
	121	100	7.8 ^h^	7.4 ^l^	17.4 ^i^	6.2 ^jk^	13.3 ^ij^	1.4 ^l^
	121	1000	5.3 ^h^	7.1 ^l^	11.1 ^jk^	2.1 ^k^	10.9 ^j^	0.9 ^l^
SEM_pooled_ *			0.093	0.125	0.115	0.145	0.085	0.098

Different lowercase letter superscripts indicate a significant difference among the means within a column (*p* < 0.05). * Pooled Standard Error of Mean.

**Table 5 foods-13-01517-t005:** Total percentage areas of different secondary structures in Amide I region of proteins milk dispersions with pH adjusted to 6.1, 6.4, 6.8, or 7.5 and subjected to different temperatures (20, 90, or 121 °C) and shear rates (0, 100, or 1000 s^−1^).

pH	Temp (°C)	Intramolecular β-Sheets (1615–1637)			Random Coils (1638–1645)			α-Helix (1646–1664)			β-Turns (1665–1681)			Aggregated β-Sheets (1682–1700)		
		0 s^−1^	100 s^−1^	1000 s^−1^	0 s^−1^	100 s^−1^	1000 s^−1^	0 s^−1^	100 s^−1^	1000 s^−1^	0 s^−1^	100 s^−1^	1000 s^−1^	0 s^−1^	100 s^−1^	1000 s^−1^
6.1	20	31.85 ^EF^			4.31 ^F^			18.31 ^E^			35.14 ^C^			10.39 ^B^		
	90	28.70 ^Ga^	21.97 ^Eb^	18.72 ^Db^	5.11 ^Fc^	10.35 ^Fb^	12.81 ^Ga^	15.49 ^Fa^	10.51 ^Eb^	6.53 ^Dc^	37.19 ^Bc^	38.83 ^Bb^	41.81 ^Ba^	13.51 ^Ac^	18.34 ^Ab^	20.13 ^Ba^
	121	23.01 ^Ha^	19.88 ^Fb^	10.94 ^Eb^	5.25 ^Fc^	6.88 ^Gb^	11.53 ^Ga^	10.51 ^Ga^	6.82 ^Fb^	5.99 ^Db^	47.65 ^Aa^	48.55 ^Aa^	49.01 ^Aa^	13.58 ^Ac^	17.87 ^Ab^	22.53 ^Aa^
6.4	20	36.71 ^C^			4.17 ^F^			21.15 ^CD^			29.14 ^E^			8.83 ^C^		
	90	33.11 ^DEa^	29.42 ^Cb^	24.43 ^Cc^	5.15 ^Fc^	11.11 ^Fb^	18.35 ^Fa^	17.30 ^Ea^	12.81 ^Db^	10.30 ^Cc^	33.93 ^Da^	32.53 ^Da^	30.81 ^Cb^	10.51 ^Bc^	14.13 ^Bb^	16.11 ^Ca^
	121	30.56 ^Fa^	27.44 ^Db^	24.32 ^Cc^	7.28 ^Ec^	14.28 ^Eb^	22.38 ^Ea^	11.53 ^Ga^	10.83 ^Eab^	9.95 ^Cc^	37.80 ^Ba^	35.21 ^Cb^	30.82 ^Dc^	12.83 ^Aa^	12.24 ^Ca^	12.53 ^Da^
6.8	20	45.01 ^A^			5.22 ^F^			27.01 ^A^			19.21 ^H^			3.55 ^G^		
	90	40.77 ^Ba^	27.40 ^Db^	24.19 ^Cc^	8.31 ^Ec^	20.15 ^Db^	27.18 ^Da^	21.59 ^Ca^	20.31 ^Aa^	14.83 ^ABb^	23.50 ^Gc^	27.32 ^Eb^	30.52 ^Ca^	5.83 D^Ea^	4.82 ^Fab^	3.28 ^Fb^
	121	36.07 ^Ca^	14.96 ^Gb^	11.53 ^Ec^	19.65 ^Cc^	22.51 ^Cb^	30.04 ^Ca^	10.31 ^Gb^	18.51 ^Ba^	11.35 ^Cb^	27.0 ^Fc^	38.51 ^Bb^	42.81 ^Ba^	6.97 ^Da^	5.51 ^Fab^	4.27 ^Fb^
7.5	20	40.04 ^B^			12.54 ^D^			25.51 ^B^			17.38 ^I^			4.53 E^F^		
	90	37.53 ^Ca^	33.28 ^Ab^	30.85 ^Ac^	22.28 ^Bc^	28.23 ^Bb^	38.01 ^Aa^	22.57 ^Ca^	20.11 ^Ab^	15.32 ^Ac^	12.51 ^Ja^	10.53 ^Fb^	5.51 ^Dc^	5.11 ^Ec^	7.85 ^Db^	10.31 ^Ea^
	121	34.45 ^Da^	31.52 ^Bb^	28.85 ^Bc^	28.51 ^Ac^	32.31 ^Ab^	36.21 ^Ba^	19.91 ^Da^	16.83 ^Cb^	13.58 ^Bc^	10.30 ^Ka^	8.81 ^Gb^	6.15 ^Dc^	6.83 ^Dc^	10.53 ^Db^	15.21 ^Ca^
SEM_pooled_ *		0.667			0.782			0.531			0.634			0.367		

Different uppercase letter superscripts indicate a significant difference among the means within a column, while lowercase superscript signifies a significant difference among the means within a raw; *p* < 0.05. * Pooled Standard Error of Mean.

## Data Availability

The original contributions presented in the study are included in the article, further inquiries can be directed to the corresponding author.

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
