# Peer review of "Effect of pH and Shear on Heat-Induced Changes in Milk Protein Concentrate Suspensions"

_foods, 2024, doi:10.3390/foods13101517_

Round 1
Reviewer 1 Report
Comments and Suggestions for Authors
The authors proposed investigated the effect of pH, shear, on Heat-Induced Changes in Milk Protein Concentrate Suspensions. Overall, the paper was well written and contains novel and interesting findings that will be of interest to followers of Foods. The introduction section was well constructed, as it provided adequate literature review on major concepts pertaining to milk protein concentrate, pH, and heat, which are critical to the study. The objectives were adequately presented. The chosen sample and methods were suitable in achieving reliable and valid research outcomes. Results were clearly presented and meaningful. The results were well discussed with existing literature, indicating that the authors had suitable mastering of the subject. However, some major and minor issues pertaining to the overall research and its presentation were found which should be addressed by the authors:
1.Abstract
Consider revising the abstract to emphasize the novelty of the article, possibly condensing the scientific problem into a succinct statement.
2.Keywords
The title and keywords selected by the authors are commendable. Nonetheless, to enhance the article's discoverability post-publication, it's advisable to avoid redundancy between the title and keywords. Therefore, consider replacing common words between the title and keywords with alternative relevant terms to optimize searchability.
3.Introduction
The author's preface is well written, but it is not difficult to see that previous studies have already produced some relevant research conclusions, and the author should explain the innovation of this paper?
The novelty of your work should be more highlighted.
4. Results and discussion
This part is well written, the author explains in depth and discusses the effects of pH, temperature, and shear on proteins. However, why is there no error value in Table 1-5? Is there a significance analysis?
5. Conclusion (make it Conclusions)
What is your conclusion in relation to conformational properties of proteins…? Please add this section.
The author selected pH values of 6.1, 6.4, 6.8, and 7.5 as the research objects. However, no summary of pH 6.8 samples was found in the abstract and conclusion, so it is recommended to improve them.
6. References
Note that the format of the references is consistent, such as the 25th reference.
Author Response
|
Editor’s/Reviewer’s comments |
Authors’ Response – Details of Amendments to Manuscript |
Line no. / Font Colour |
|
|
|
|
|
Thank you. We appreciate your recognition of our study and we highly value your comments and feedback for improving our manuscript. |
|
Consider revising the abstract to emphasize the novelty of the article, possibly condensing the scientific problem into a succinct statement. |
Thanks, the abstract has now been revised to emphasize the novelty of the article. |
L14-16 Additional information was added to the abstract as follows to emphasize the novelty “The effect of shear on heat-induced changes of milk protein concentrate suspensions were examined at different pH levels, revealing novel insights into micellar dissociation and protein aggregation dynamics.” |
The title and keywords selected by the authors are commendable. Nonetheless, to enhance the article's discoverability post-publication, it's advisable to avoid redundancy between the title and keywords. Therefore, consider replacing common words between the title and keywords with alternative relevant terms to optimize searchability. |
Authors do agree with the reviewer and common words between the title and keywords were replaced with some alternative terms. |
L29 Keywords were replaced as follows; Dairy, Shear effect, Temperature, pH variation |
The author's preface is well written, but it is not difficult to see that previous studies have already produced some relevant research conclusions, and the author should explain the innovation of this paper? The novelty of your work should be more highlighted. |
Introduction has now been revised to highlight the novelty of the paper as suggested. |
L71-74 Additional information was added in the text for further clarification as follows; “While previous research has explored the effects of pH and shear on MPCs, there remains a significant gap in the literature regarding the combined influence of these factors under heat treatment, particularly at the pH levels commonly encountered in various dairy applications ranging from pH 6.1 to 7.5.” |
This part is well written, the author explains in depth and discusses the effects of pH, temperature, and shear on proteins. However, why is there no error value in Table 1-5? Is there a significance analysis? |
Thank you for your feedback and inquiry regarding Table 1-5 in our paper.
The tables actually contain error terms expressed as pooled standard error of the mean (SEMpooled). This is due to use of a general linear model and split plot design which ‘pools’ all the variance from all the samples and divide it across the main and split plot factors. By pooling the SEMpooled values, we aimed to provide a concise representation of the variability in our data while ensuring clarity and readability in the table. The significance analysis conducted as part of our statistical analysis further supports our findings. We appreciate your attention to detail and clarification on this matter.
We have made improvements in the description of the statistical analysis to explain this better |
L135-137 Statistical analysis method was improved as follows; “Statistical analysis was conducted using IBM SPSS Statistics software (version 28.0.1.0, IBM Corp., Armonk, NY) employing a general linear model (GLM) approach. The study was arranged as a randomised block, full factorial design with pH as the main plot and, while the subplots were temperature/time combinations and shearing. This block was replicated at least twice, with each replication consisting of three sub-samplings. The level of significance was set at P ≤ 0.05.” |
What is your conclusion in relation to conformational properties of proteins…?
|
Conclusions have now been revised to specify the conformational properties of proteins at different pH levels under combined heat and shear. |
Information regarding conformational properties was added in the first paragraph of conclusions section as follows. L443-450 – “A shear-dependent reduction in intramolecular β-sheets and α-helical content was observed at both temperatures at both pH levels suggesting alterations in protein conformation. Shear-induced fragmentation of protein aggregates was clearly noticeable in the control MPC suspensions at pH 6.8, regardless of the heating temperature. The combination of heat and shear applied to MPC suspensions at pH 6.8 resulted in a reduction of intramolecular β-sheet structures and an increase in β-turns. There was also a decrease in α-helix content and an increase in random coils. Additionally, aggregated β-sheet structures decreased compared to heated dispersions.” L453-455 - At pH 7.5, the combined application of heat and shear decreased the content of α-helix structures compared to heated suspensions due to changes in protein structure.”
|
|
Summary of pH 6.8 samples was added to abstract, and conclusions as recommended. |
L18-19 – “The fragmentation of protein aggregates induced by shear was evident in the control MPC suspensions at pH 6.8, irrespective of the temperature” L445-448 – “Shear-induced fragmentation of protein aggregates was clearly noticeable in the control MPC suspensions at pH 6.8, regardless of the heating temperature. The combination of heat and shear applied to MPC suspensions at pH 6.8 resulted in a reduction of intra-molecular β-sheet structures and an increase in β-turns. There was also a decrease in α-helix content and an increase in random coils. Additionally, aggregated β-sheet structures decreased compared to heated dispersions” |
Note that the format of the references is consistent, such as the 25th reference. |
Corrected as suggested. |
L529 - reference was corrected as follows; Aydogdu, T. Investigations of the Complex Relationships between Minerals, pH, and Heat Stability in Milk Protein Systems. PhD Thesis, University College Cork, Cork, Ireland, 2023. |
Reviewer 2 Report
Comments and Suggestions for Authors
This manuscript entitled "Effect of pH and Shear on Heat-Induced Changes in Milk Protein Concentrate Suspensions" by Anushka Mediwaththe and co-authors delves into the effects of heat and shear on milk protein concentrate suspensions. The study reveals that the behavior of the suspensions is pH-dependent. Higher pH levels were found to cause micellar dissociation under intense conditions. Conversely, lower pH levels showed diverse outcomes in aggregation and fragmentation, which were influenced by electrostatic interactions. Nonetheless, there are a few issues with the manuscript that require attention before it can be accepted for publication. Some suggestions are provided below to improve the manuscript quality.
1. In the introduction, the author describes a noticeable decrease in the aggregation size of casein under low pH conditions (such as pH 2.0 and pH 4.6). However, the four pH conditions the author chose do not include lower pH values. Please explain the reason for selecting these four pH conditions.
2. In Figure 1, please change the label from '90C/0' to '90°C/0s,' and make the same modification in the other figures.
3. Please explain the meaning of the superscripted letters for each value in Tables 1 and 2.
4. Section 2.4: Please explain the method used to calculate the percentage of secondary structure from FTIR experimental data.
5. Line 366: Describe the relationship between the pH of the solution and the isoelectric point of proteins. Please describe the correlation between the pI value of casein and the pH of the solution.
6. Line 117: Please provide the full names of αS1-CN, αS2-CN, β-CN, κ-CN, α-LA, and β-LG to enhance the readability of the article.
Author Response
|
Editor’s/Reviewer’s comments |
Authors’ Response – Details of Amendments to Manuscript |
Line no. / Font Colour |
|
|
|
|
|
This manuscript entitled "Effect of pH and Shear on Heat-Induced Changes in Milk Protein Concentrate Suspensions" by Anushka Mediwaththe and co-authors delves into the effects of heat and shear on milk protein concentrate suspensions. The study reveals that the behavior of the suspensions is pH-dependent. Higher pH levels were found to cause micellar dissociation under intense conditions. Conversely, lower pH levels showed diverse outcomes in aggregation and fragmentation, which were influenced by electrostatic interactions. Nonetheless, there are a few issues with the manuscript that require attention before it can be accepted for publication. Some suggestions are provided below to improve the manuscript quality. |
Thank you. We greatly appreciate your feedback to improve our manuscript. |
|
|
1. In the introduction, the author describes a noticeable decrease in the aggregation size of casein under low pH conditions (such as pH 2.0 and pH 4.6). However, the four pH conditions the author chose do not include lower pH values. Please explain the reason for selecting these four pH conditions. |
Introduction has now been revised to highlight the novelty of the paper including the reason for selecting four pH conditions. |
L71-74 – “While previous research has explored the effects of pH and shear on MPCs, there remains a significant gap in the literature regarding the combined influence of these factors during heat treatment, particularly at the pH levels commonly encountered in various dairy applications ranging from pH 6.1 to 7.5.” |
|
2. In Figure 1, please change the label from '90C/0' to '90°C/0s,' and make the same modification in the other figures. |
We appreciate your suggestion regarding updating the figure labels. However, considering the clarity and consistency within the figures, we have opted to retain the current figure labels. Instead, we have updated the captions as per your recommendation to reflect the desired format as follows; “Please note that in the figure labels, 'C' represents degrees Celsius (℃), and the numerical value following the slash denotes per second (s-1).” We believe this approach maintains the integrity of the figures while addressing your concern. Thank you for your understanding. |
|
|
3. Please explain the meaning of the superscripted letters for each value in Tables 1 and 2. |
A detailed description regarding the superscript letters were provided below the table 1 and 2. |
L173-175 & L217-218 - “The means in a column with different superscript uppercase letters and in a row with different superscript lowercase letters differ significantly (p<0.05). *Pooled Standard Error of Mean.” |
|
4. Section 2.4: Please explain the method used to calculate the percentage of secondary structure from FTIR experimental data. |
Method in section 2.4 has been revised as recommended. |
L127-139 – “FTIR spectra were acquired at room temperature (~20 ℃) within 10 minutes after each treatment. Each spectrum was an average of 16 scans with a resolution of 4 cm-1 after subtracting the background. To enhance resolution for qualitative analysis, the second derivative of all FTIR spectra was obtained within the broad amide I region of 1700-1600 cm -1. The obtained spectra were processed using Fourier self-deconvolution (FSD) and baseline correction with Origin Student 2019b software (Origin Lab Corporation, Northampton, MA, USA). The areas of the prominent peaks assigned to specific secondary structures were summed up and divided by the total area, resulting in the identification of five major peak areas corresponding to protein secondary structures, including intramolecular β sheets (1637-1615 cm-1), aggregated β sheets (1700-1682 cm-1), random coils (1645-1638 cm-1), α-helices (1664-1646 cm-1), and β turns (1681-1665 cm-1). The obtained results were then subjected to statistical analysis following the guidelines outlined in section 2.6.” |
|
5. Line 366: Describe the relationship between the pH of the solution and the isoelectric point of proteins. Please describe the correlation between the pI value of casein and the pH of the solution. |
This section has been revised as advised. |
L398-403 – “The pH of a solution significantly affects the protein behaviour, particularly through its relation to the protein's isoelectric point (pI). The pI represents the pH at which a protein carries no net charge. For casein, with a pI around 4.6, proximity of the solution's pH to the pI influences solubility and conformational changes, affecting interactions with other molecules. This correlation between casein's pI and solution pH directly affects micellar stability and aggregation tendencies [11].” |
|
6. Line 117: Please provide the full names of αS1-CN, αS2-CN, β-CN, κ-CN, α-LA, and β-LG to enhance the readability of the article. |
Full names of proteins were provided as recommended. |
L142-146 – “Whole samples and supernatants of each centrifuged MPC suspensions were analysed for the content of αS1-Casein (αS1-CN), αS2-Casein (αS2-CN), β-Casein (β-CN), κ-Casein (κ-CN), α-lactalbumin (α-LA) and β-lactoglobulin (β-LG) by RP-HPLC using a Zorbax 300SB-C8 RP-HPLC column (silica-based packing, 3.5 micron, 300A, Agilent Technologies Inc., Mulgrave VIC, Australia) as the stationary phase.” |
Reviewer 3 Report
Comments and Suggestions for Authors
Dear Authors,
The manuscript clearly described differences between a few differently treated samples of MPC. Technology is not new but maybe with a new perspective. I have two main concerns:
- Statistics you use (for instance Tab. 1 and 2 present only one SEM value but samples and treatments were a lot); Which group did it describe and did you properly analyse the data?
- You did not present an idea for what those findings could be used. The sentence "Understanding these interactions is crucial for optimizing processing conditions and product properties in various dairy applications" is just a slogan not proper for research findings.
Please complete this manuscript to make it more interesting for readers.

Author Response
|
Editor’s/Reviewer’s comments |
Authors’ Response – Details of Amendments to Manuscript |
Line no. / Font Colour |
|
|
|
|
|
Thank you. We greatly value your feedback and suggestions to enhance our manuscript. |
|
|
The data has been properly analysed and we addressed this in the comment made to reviewer #1. We made further refinements to clarify this. Use of SEMpooled is yet another way to deal with variance and establish if there is a significant difference among the samples. We have added additional clarification in the statistical method description. |
L135-137 Statistical analysis method was improved as follows; “Statistical analysis was conducted using IBM SPSS Statistics software (version 28.0.1.0, IBM Corp., Armonk, NY) employing a general linear model (GLM) approach. The study was arranged as a randomised block, full factorial design with pH as the main plot and, while the subplots were temperature/time combinations and shearing. This block was replicated at least twice, with each replication consisting of three sub-samplings. The level of significance was set at P ≤ 0.05.” |
|
This sentence was revised as recommended. |
L457-460 - Information was added in the conclusions section as follows for further clarification; “This understanding aids dairy manufacturers in optimizing processing conditions for improved product stability, texture, and functionality. It also supports the development of processing techniques that preserve protein functionality and nutritional integrity in dairy products.” |
|
Thank you for your feedback and inquiry regarding Table 1-5 in our paper.
The tables actually contain error terms expressed as pooled standard error of the mean (SEMpooled). This is due to use of a general linear model and split plot design which ‘pools’ all the variance from all the samples and divide it across the main and split plot factors. By pooling the SEMpooled values, we aimed to provide a concise representation of the variability in our data while ensuring clarity and readability in the table. The significance analysis conducted as part of our statistical analysis further supports our findings. We appreciate your attention to detail and clarification on this matter.
We have made improvements in the description of the statistical analysis to explain this better |
L135-137 Statistical analysis method was improved as follows; “Statistical analysis was conducted using IBM SPSS Statistics software (version 28.0.1.0, IBM Corp., Armonk, NY) employing a general linear model (GLM) approach. The study was arranged as a randomised block, full factorial design with pH as the main plot and, while the subplots were temperature/time combinations and shearing. This block was replicated at least twice, with each replication consisting of three sub-samplings. The level of significance was set at P ≤ 0.05.” |
|
Corrected as suggested. |
L152-154 – corrected as follows; “The means in a column with different superscript uppercase letters and in a row with different superscript lowercase letters differ significantly (p<0.05). *Pooled Standard Error of Mean.” |
|
Heating under control pH conditions (pH 6.8) results in whey protein denaturation and particle sizes were illustrated in the Table 1. However, the observed increase in particle size at pH 6.1 and 6.4, as illustrated in Table 1, surpasses what would be expected solely from whey protein denaturation. This suggests the casein micelle aggregation, influencing particle size changes. Additional references on whey protein denaturation at similar temperatures to MPC but without casein micelle aggregation were added within the text for further clarification as follows; 1. Mediwaththe, A., Huppertz, T., Chandrapala, J., & Vasiljevic, T. (2024). Effect of protein content on heat stability of reconstituted milk protein concentrate under controlled shearing. Foods, 13(2), 263. 2. Crowley, S. V. Physicochemical characterisation of protein ingredients prepared from milk by ultrafiltration or microfiltration for application in formulated nutritional products. PhD Thesis, University College Cork, 2016. |
L156-159 – The sentence was revised as follows; “Heating the MPC suspensions with initial pH values of 6.1 or 6.4 led to an increase in particle size and these were larger at pH 6.1 than at pH 6.4 (Figure 1 and Table 1); the extent of these increases was far larger than what would be expected from whey protein denaturation [16,17] and suggests it to be due to casein micelle aggregation.” |
|
It appears that there is some misunderstanding here so we have modified the statement |
L156-159 Heating the MPC suspensions with initial pH values of 6.1 or 6.4 led to an increase in particle size and these were larger at pH 6.1 than at pH 6.4 (Figure 1 and Table 1); the ex-tent of these increases was far larger than what would be expected from whey protein denaturation [16,17] and suggests it to be due to casein micelle aggregation. |
|
Thank you for bringing this to attention. The authors completely agree that there was a data misinterpretation, which has now been corrected. |
L173-176 – corrected as follows; “At pH 6.8, combined application of heat and shear did not change the particle size. However, at pH 7.5, a consistent trend of particle size reduction under the influence of heat and shear was observed, with the smallest particle size observed at a shear rate of 1000 s-1 at 121 ℃.” |
|
Please refer to response regarding comment 10. |
|
|
Revised as recommended. |
L457-460 - Information was added in the conclusions section as follows; “This understanding aids dairy manufacturers in optimizing processing conditions for improved product stability, texture, and functionality. It also supports the development of processing techniques that preserve protein functionality and nutritional integrity in dairy products.” |
Reviewer 4 Report
Comments and Suggestions for Authors
There are many areas where the explanations are inadequate prior to reviewing, and I feel that further review will be very difficult unless these areas are corrected. The following is a list of the main points that are unclear, and the reviewer require that they be corrected.
1. There is not superscripted asterisk character on author name as indicator of the corresponding author.
2. The abbreviated words "k-CN" appeared in the abstract (line 17). But, the authors did not define this abbreviated word before this word was described.
3. In line 92, why did the authors select this "constant pressure of 250 kPa"? The authors have to describe the reason for readers' understanding.
4. In the sections 2.3 and 2.4, the authors only indicated your previous paper, reference [14], and did not explain the procedure briefly. The authors should describe the procedure briefly for readers' understanding. (line 106-113 )
5. In the section 2.5, many abbreviated words were appeared, however, there were not the explanation of those abbreviated words. The authors have to explain those words.
6. In all tables, the superscripted characters, for example, “Cc” were not defined and explained in the manuscript. The authors have to define and explain those superscripted characters for readers' understanding.
Author Response
|
Editor’s/Reviewer’s comments |
Authors’ Response – Details of Amendments to Manuscript |
Line no. / Font Colour |
|
|
|
|
|
1. There are many areas where the explanations are inadequate prior to reviewing, and I feel that further review will be very difficult unless these areas are corrected. The following is a list of the main points that are unclear, and the reviewer require that they be corrected. |
Thank you. We value your feedback to improve our manuscript. |
|
|
2. There is not superscripted asterisk character on author name as indicator of the corresponding author. |
Superscripted asterisk was added to the name of the corresponding author as advised. |
|
|
3. The abbreviated words "k-CN" appeared in the abstract (line 17). But, the authors did not define this abbreviated word before this word was described. |
The word “κ-CN” was defined in the abstract as advised. |
L21 – “This effect was particularly pronounced at 121 ℃ and 1000 s-1, resulting in reduced particle size and an elevated concentration of κ-casein (κ-CN) in the non-sedimentable phase.” |
|
4. In line 92, why did the authors select this "constant pressure of 250 kPa"? The authors have to describe the reason for readers' understanding. |
Information was added in the text for further clarification. |
L100-103 – “This pressure is crucial for eliminating air bubbles, enhancing sample-to-measuring surface contact, stabilizing the sample during testing, ensuring consistent experimental conditions, and preventing evaporation when heating the samples, thereby facilitating accurate and reproducible rheological measurements across various samples.” |
|
5. In the sections 2.3 and 2.4, the authors only indicated your previous paper, reference [14], and did not explain the procedure briefly. The authors should describe the procedure briefly for readers' understanding. (line 106-113 ) |
Procedures in section 2.3 and 2.4 were briefly explained as recommended. |
Section 2.3 – L-118-122 – “The measurements were performed using a Zetasizer (Zetasizer Nano ZS, Malvern Instruments, Malvern, UK). Prior to the measurements, the treated samples were diluted 1000 times using skim milk ultra-filtrate. In the calculations, the refractive indexes of 1.57 and 1.38 were used for MPC and skim milk ultra-filtrate, respectively.” Section 2.4 – L127-139 – “FTIR spectra were acquired at room temperature (~20 ℃) within 10 minutes after each treatment. Each spectrum was an average of 16 scans with a resolution of 4 cm-1 after subtracting the background. To enhance resolution for qualitative analysis, the second derivative of all FTIR spectra was obtained within the broad amide I region of 1700-1600 cm -1. The obtained spectra were processed using Fourier self-deconvolution (FSD) and baseline correction with Origin Student 2019b software (Origin Lab Corporation, Northampton, MA, USA). The areas of the prominent peaks assigned to specific secondary structures were summed up and divided by the total area, resulting in the identification of five major peak areas corresponding to protein secondary structures, including intramolecular β sheets (1637-1615 cm-1), aggregated β sheets (1700-1682 cm-1), random coils (1645-1638 cm-1), α-helices (1664-1646 cm-1), and β turns (1681-1665 cm-1). The obtained results were then subjected to statistical analysis following the guidelines outlined in section 2.6.” |
|
6. In the section 2.5, many abbreviated words were appeared, however, there were not the explanation of those abbreviated words. The authors have to explain those words. |
Abbreviated words within section 2.5 were explained within the text as recommended. |
L142-146 - Whole samples and supernatants of each centrifuged MPC suspensions were analysed for the content of αS1-casein (αS1-CN), αS2-casein (αS2-CN), β-casein (β-CN), κ-casein (κ-CN), α-lactalbumin (α-LA) and β-lactoglobulin (β-LG) by RP-HPLC using a Zorbax 300SB-C8 RP-HPLC column (silica-based packing, 3.5 micron, 300A, Agilent Technologies Inc., Mulgrave VIC, Australia) as the stationary phase. |
|
7. In all tables, the superscripted characters, for example, “Cc” were not defined and explained in the manuscript. The authors have to define and explain those superscripted characters for readers' understanding |
In all tables, superscripted characters were clearly defined under the tables (as a footnote). |
|
Round 2
Reviewer 1 Report
Comments and Suggestions for Authors
After revision, the manuscript can be published in this journal.
Comments on the Quality of English LanguageNONE
Author Response
Thank you for your kind words.
Reviewer 4 Report
Comments and Suggestions for Authors
My previous comment 7, which I don't think has been corrected. I have seen the revised manuscript, and there is no explanation for the individual superscripts. For example, in Table 1, there are E, Cc, Bb, Ba, Ac, Ab, Aa, F, Da, Dc, Db, H, Ja, Ea, Ia, Eb, Fc, Fb, and I have no idea what they mean. I can point out exactly the same thing in all the tables. I think a more detailed explanation is necessary for the reader's understanding. This is the last opportunity to correct the manuscript.
Author Response
We appologise if this wasn't clear in the text. These upper and lowercase letters indicate the significant difference among the means within a column or a raw in the associated table. We rephrased the text so hopefully it is now clearer.